# NeuroDante: Poetry Mentally Engages More Experts but Moves More Non-Experts, and for Both the Cerebral Approach Tendency Goes Hand in Hand with the Cerebral Effort

**DOI:** 10.3390/brainsci11030281

**Published:** 2021-02-25

**Authors:** Giulia Cartocci, Dario Rossi, Enrica Modica, Anton Giulio Maglione, Ana C. Martinez Levy, Patrizia Cherubino, Paolo Canettieri, Mariella Combi, Roberto Rea, Luca Gatti, Fabio Babiloni

**Affiliations:** 1Department of Molecular Medicine, University of Rome Sapienza, Viale Regina Elena 291, 00161 Rome, Italy; dario.rossi@uniroma1.it (D.R.); enrica.modica@uniroma1.it (E.M.); antongiulio.maglione@uniroma1.it (A.G.M.); ana.martinezlevy@uniroma1.it (A.C.M.L.); patrizia.cherubino@uniroma1.it (P.C.); paolo.canettieri@uniroma1.it (P.C.); mariella.combi@uniroma1.it (M.C.); roberto.rea@uniroma1.it (R.R.); luca.gatti@uniroma1.it (L.G.); fabio.babiloni@uniroma1.it (F.B.); 2BrainSigns Srl, Lungotevere Michelangelo 9, 00198 Rome, Italy; 3Department of Business and Management, LUISS Guido Carli University, Viale Romania 32, 00197 Roma, Italy; 4College Computer Science and Technology, University Hangzhou Dianzi, Hangzhou 310018, China

**Keywords:** EEG, alpha, theta, frontal theta, frontal alpha asymmetry, skin conductance response, neuroaesthetics, neurocognitive poetics, time, Divina Commedia

## Abstract

Neuroaesthetics, the science studying the biological underpinnings of aesthetic experience, recently extended its area of investigation to literary art; this was the humus where neurocognitive poetics blossomed. Divina Commedia represents one of the most important, famous and studied poems worldwide. Poetry stimuli are characterized by elements (meter and rhyme) promoting the processing fluency, a core aspect of neuroaesthetics theories. In addition, given the evidence of different neurophysiological reactions between experts and non-experts in response to artistic stimuli, the aim of the present study was to investigate, in poetry, a different neurophysiological cognitive and emotional reaction between Literature (L) and Non-Literature (NL) students. A further aim was to investigate whether neurophysiological underpinnings would support explanation of behavioral data. Investigation methods employed: self-report assessments (recognition, appreciation, content recall) and neurophysiological indexes (approach/withdrawal (AW), cerebral effort (CE) and galvanic skin response (GSR)). The main behavioral results, according to fluency theories in aesthetics, suggested in the NL but not in the L group that the appreciation/liking went hand by hand with the self-declared recognition and with the content recall. The main neurophysiological results were: (i) higher galvanic skin response in NL, whilst higher CE values in L; (ii) a positive correlation between AW and CE indexes in both groups. The present results extended previous evidence relative to figurative art also to auditory poetry stimuli, suggesting an emotional attenuation “expertise-specific” showed by experts, but increased cognitive processing in response to the stimuli.

## 1. Introduction

Neuroaesthetics is a growing field of research investigating the neurobiological correlations of the aesthetic experience [1,2]. Such an area of investigation is more classically focused on figurative arts, but it has been widening its areas of interest to text material, as witnessed by the *neurocognitive poetics*, which saw its dawn around 10 years ago and is defined as “the transdisciplinary empirical investigation of and theorizing about (poetic) literature reception by eye or ear including its neuronal underpinnings” [3]. The rationality and validity of such an area of investigation are clearly summarized into the sentence expressed some years earlier by Turner and Pöppel [4]: ‘‘Poetry presents to the brain a system which is temporally and rhythmically hierarchical, as well as linguistically so, and therefore matched to the hierarchical organization of the brain itself’’. A core element of the neurocognitive poetic model [3] is the *fiction feeling hypothesis*. Such hypothesis was born from the evidence that the medial and bilateral orbitofrontal cortex activation (OFC), known to be involved in social cognition, was engaged when processing stories with an emotional content, stories that would then elicit affective and cognitive empathy in the reader [5]. In addition, the medial part of the OFC has been found to be active during the exposure to both audio and visual beautiful stimuli [6]. However, higher-level cognitive processes, such as comprehension and reasoning, that are crucial for texts processing, are related to the prefrontal cortex activity (for a review [7]); therefore, in the present study, the entire prefrontal area was considered, not limiting the investigation to its most anterior part, that is the OFC. Moreover, in addition to studies focusing on subcortical regions’ (amygdala and insula) activation in correspondence with emotional engagement in response to words [8,9], in a study investigating the effect of the emotional valence on words processing, an activation of the bilateral inferior frontal gyrus in response to emotional words was found, and, in the superior frontal gyrus, is already known to support semantic retrieval, in response to positive words, therefore suggesting an association between familiarity and positive feelings for verbal material [10]. In fact, recently, researchers have started to measure cerebral and emotional activities, manipulating some factors that could alter or increase pleasure in the story perception [11,12]. In particular, an increase in liking and rhythmicity rating for metered and rhyming stanzas was found, which, from a neurophysiological point of observation, elicited smaller event-related potential (ERP) responses (N400/P600) in comparison to nonmetered, nonrhyming, or nonmetered and nonrhyming counterparts [12]. Such results are in accord with the *cognitive fluency theory* [13], predicting that higher perceptual fluency and, consequently, a higher aesthetic appreciation, would be produced by recurring patterns, such as meter. 

In addition to ERP, an electroencephalographic EEG cerebral index of approach or withdrawal (AW) motivation toward a stimulus, suggesting an appreciation or avoidance toward it is based on the relative alpha asymmetry in the frontal area [14,15]. According to the AW model, a relative power suppression of the alpha rhythm across the left frontal cortex is associated with a propensity to approach a stimulus, while the relative power suppression of the alpha rhythm across the right frontal cortex is associated with a propensity to withdraw from a stimulus [16]. The AW index has been already applied to several kinds of audio–visual material, such as musicals [17,18], public service announcements [19,20,21,22,23], foreign products [24], commercials [25,26,27,28,29,30] and, importantly, neuroaesthetics, in response to the observation of: the real sculpture of Michelangelo’s Moses, XVII century Dutch painters and Titian masterpieces [31,32,33]. Concerning literature, it was applied to the study of the reaction to the reading of a novel [34]. Moreover, it was already employed in a preliminary investigation concerning the same poetic excerpts adopted in the present study, evidencing higher approach tendency in experts than in non-experts on Classical Literature [35]. The expertise on a certain topic has been identified as a key factor for the execution and the reaction to several kinds of stimuli, reflected not only by behavioral outcomes but also by neurophysiological parameter correlations. This was found in air traffic controllers for frontal and parietal theta synchronization and frontal alpha desynchronization [36,37], in wine sommeliers for frontal alpha asymmetry and brain activity estimated by fMRI [38,39], in pilots and Air Traffic Management for frontal theta and parietal alpha [40], in professional shooters for midline frontal theta [41], in dancers for alpha power desynchronization distributed over the scalp [42], in dancers for theta and gamma synchrony [43], in musicians for alpha and beta synchrony [43], and in art experts for approximate entropy [44]. Concerning the last application, that is the investigation of neural correlations elicited in experts and non-experts in response to artworks, only a few studies have been conducted till now (see below), despite several pieces of evidence being reported that employ behavioral approaches such as ratings of liking and understanding of visual art [45,46,47,48] (Table 1).

On the contrary, scientific literature presents a plethora of studies investigating the brain activity in response to music listening in expert and non-expert musicians [49,50,51,52,53].

In addition to EEG indexes, the suitability of autonomic indexes for the study of emotional response is well-known [55]. Several studies and models support the notion that emotions play an important role in making aesthetic judgments [56,57]; therefore, since the involvement of arousal alteration is present in all kinds of emotional reaction [58], in the present study it has been focused on such dimension in relation to neurophysiological autonomic indexes. Several studies evidenced that, among them, the galvanic skin response (GSR) is strictly related to the rated arousal of emotional stimuli (e.g., [59,60,61]). Arousal represents one of the two dimensions of emotions, together with valence [62], and as GSR variations, it has already been employed in the study of emotion applied to different fields such as products and advertisement [20,22,24,27], music [63] and, obviously, art [31].

To the scope of the present study, it is possible to consider the poetry made up by the sum of musical properties (constituted by the meter and the rhythm) and of a narrative content, while the relative paraphrase is constituted by the sole content. Given that we subtracted the brain activity estimated in relation to the poetry being listened to by the brain activity estimated in relation to paraphrase listening, and then divided by the sum of both the activities Equation (3), the result would provide positive values whether the higher relative activity occurred during poetry listening, and negative results whether the higher relative activity occurred during paraphrase listening. Employing such index, the investigation will be performed regarding different neurophysiological patterns in experts and non-experts in relation to the eventual modulation of the cognitive and emotional response to poetry or paraphrase stimuli.

The famous Italian XIV century poem the “Divina Commedia” by Dante Alighieri (1265–1321) is characterized by a repetitive and constant structure, as it is composed by three parts (cantiche: Inferno—Hell; Purgatorio—Purgatory; Paradiso—Paradise), each part is composed by thirty-three cantos, for an average length of 142 verses. The verse scheme used, “terza rima”, is hendecasyllabic (lines of eleven syllables), with the lines composing tercets relying on the rhyme scheme xyx yzy z. Each cantica takes place in a different environment, as suggested by their titles, so the reader follows Dante’s journey through Hell, Purgatory and Paradise, where the poet, respectively, meets damned souls, then souls expiating their sins and finally enjoys the vision of God. Studying the comparison among different cantica would enable one to assess the effect of a potentially differential previous knowledge of the peculiar cantica. Furthermore, the reaction to the exposure to stanzas belonging to each of the three cantiche would enable one to test the effect of the content of the excerpts, maintaining the meter and the rhyme scheme constant.

Experimental Objectives:Poetry stimuli are characterized by elements (meter and rhyme) promoting the processing fluency, a core aspect of neuroaesthetics theories. In addition, given the evidence of a different neurophysiological reaction between experts and non-experts in response to artistic stimuli, in light of the possible generalizability of neuroaesthetic concepts [64], the aim of the present study was to investigate in the literature art, and specifically in poetry, eventual differences in the neurophysiological cognitive and emotional reaction between the Literature (L) and Non-Literature (NL) students groups enrolled in the study.Since traditional approaches employ declarative behavioral data, such as appreciation or comprehension rating [45,46,47,48], and it has been proved that self-report data could be affected by many confounding factors [65,66,67], a further aim of the present study was to investigate whether neurophysiological underpinnings could support explanation of behavioral data.

## 2. Materials and Methods

In the present study, 47 healthy participants (23 Literature students; 12 females, 11 males; mean age 25.391 ± 4.408, and 24 Non-Literature students; 12 females, 12 males; mean age 26.667 ± 2.316 years old) have been enrolled on a voluntary base; they have not received any compensation from taking part in the research. Participants were university students at Sapienza University of Rome, Literature (L) students attending humanistic courses and the second half scientific courses. All participants were given detailed information on the study and signed an informed consent. The experiment was performed in accord with the principles outlined in the Declaration of Helsinki of 1975, as revised in 2000, and it was approved by the university’s ethical committee.

The project identification code was RM11916B5ADDCB0B, as assigned by the University of Rome Sapienza on 21 April 2016.

Participants were sitting on a comfortable chair and instructed to listen to the auditory stimuli that were delivered through earphones at approximately 65 dB SPL [68,69,70], operating slight changes in order to fit each participant with a level of comfortable intensity. The stimuli were constituted by three excerpts from the reading of Dante Alighieri’s Divina Commedia (see Appendix A) and the corresponding paraphrase. The three selected emblematic pieces have been chosen by experts in Italian literature (academic professors and researchers) and each belonged to one of the three cantiche of the poem: canto V from Inferno (Hell), canto XXX from Purgatorio (Purgatory) and canto XXXIII from Paradiso (Paradise), respectively. All the texts were read by an Italian professional actor to ensure the quality of the elocution. The mean duration of the read pieces was 189.833 ± 18.999 s and they were pseudo-randomly played, producing different trains of auditory stimuli in order to balance among participants the order of presentation of the three cantiche. The train was preceded and followed by sentences in Italian language that belong to a standardized set of sentences used normally for audiometric purposes in clinics [71]. Such sequence of short phrases lasted 1 minute of total length and has been used as the baseline in this experimental setup. The employed experimental protocol resembles one adopted in a previous pilot study [35] that employed almost the same neurometric indexes used in the present study: the Approach–Withdrawal index (AW) [19,20,25], the Cerebral Effort index (CE) [70,72,73] and the Emotional Index (EI) [19,20,26]. In the present study, we maintained AW and CE indexes, while, in contrast to the Emotional Index, that is a combination of the study of the heart rate and the galvanic skin response based on Russell’s circumplex model of affects [62,74], we used the GSR, as index of emotional arousal [55]. 

The EEG activity was recorded using a 10-electrodes-based EEG frontal band (Fpz, Fp1, Fp2, AFz, AF3, AF4, AF5, AF6, AF7, AF8) by means of a portable 24-channel system (BEmicro, EBneuro, Italy). The signals were acquired at a sampling rate of 256 Hz and the impedances were kept below 10 kΩ. In order to reject the main current interference, a notch filter (50 Hz) was applied, and then the gathered signal was digitally band-pass filtered by a 5th order Butterworth filter (2 ÷ 30 Hz), in order to reject the continuous component, as well as high-frequency interferences, such as muscular artifacts. Successively, in order to identify and remove other artifact-related components, such as blinks and eye movements, an independent component analysis (ICA) was applied to EEG data [75]. Furthermore, in order to take into account any subjective differences in terms of brain rhythms, for each subject the individual alpha frequency (IAF) was computed on the 60-second-long closed eyes segment [76], recorded at the beginning of the experimental task, in order to define the EEG bands of interest as: theta (IAF − 6 ÷ IAF − 2 Hz) and alpha (IAF − 2 ÷ IAF + 2 Hz) [72]. Moreover, the global field power (GFP) [77] was calculated, so as to summarize the activity of the cortical areas of interest in a specific frequency band. Specifically, in the present study, the GFP was computed from a specific set of electrodes for each index, by performing the sum of squared values of EEG potential at each electrode, averaged for the number of involved electrodes, resulting in a time-varying waveform related to the increase or decrease in the global power in the analyzed EEG. The GFP was defined according to Formula (1):(1)$$GFP_{\vartheta ,\;Frontal}\;\left(t\right)=\frac{1}{N}\sum_{i=1}^{N}x_{i,\vartheta }{\left(t\right)}^{2},$$
where ϑ is the considered EEG band, *Frontal* is the considered cortical area, *N* is the number of electrodes included in the area of interest, and *i* is the electrodes’ index. In addition, the GFP function was averaged on 1-second-long signal windows, so to comply with the EEG signal stationarity hypothesis [78].

In particular, for the AW index calculation, Formula (2) was employed [73]:AW = GFPα_right − GFPα_left,(2)
where the GFPα_right and GFPalpha_left stand for the GFP calculated among right (Fp2, AF4, AF6, AF8) and left (Fp1, AF3, AF5, AF7) electrodes, respectively, in the alpha (α) band. Positive AW values stand for a participant’s approach tendency, while negative AW values for a withdrawal tendency in relation to the stimulus.

Concerning the CE index, GFP in the theta band over all the frontal electrodes (Fpz, AFz, Fp2, AF4, AF6, AF8, AF7, AF3, Fp1, AF5) was considered for the index computation. Increased frontal theta (that is CE) values would imply an increase in the task difficulty [79].

Concerning the GSR, the electrodermal activity was recorded by means of a NeXus-10 (Mindmedia, The Netherlands) system with a sampling rate of 128 Hz and the skin conductance acquired by the constant voltage method (0.5 V). The electrodes were attached, on the non-dominant hand, to the palmar side of the middle phalanges of the second and third fingers of the participant, following published procedures [80]. The tonic component of the skin conductance level was obtained using LEDAlab software [81].

In order to assess the influence exerted by the rhythm and the meter and the specific words composition of each excerpts, overcoming an eventual modulation of the participants’ response exerted by the content, GFP and GSR (Index) data were standardized according to Formula (3) for each cantica excerpt:Index_Rel Poet_ = (Index_Poetry_ − Index_Paraphrase_) ÷ (ABS(Index_Poetry_) + ABS(Index_Paraphrase_)),(3)
where Index_Rel Poet_ stands for GFP or GSR data corresponding to the listening to the poetry version of the cantica excerpts (Index_Poetry_) relative to the GFP or GSR data corresponding to the listening to the paraphrase version of them (Index_Paraphrase_). ABS stands for absolute values of the Index_Poetry_ and Index_Paraphrase_. Positive Index_Rel Poet_ values stand for higher responses to poetry than to paraphrase, while negative Index_Rel Poet_ values stand for higher responses to paraphrase than to poetry.

At the end of the listening session, participants were asked to fill in a short-written questionnaire, investigating whether they recognized and appreciated the audio pieces (yes/no) and to say what they have heard (free written description). The written descriptions of each participant for each of the cantiche were analyzed by authors with specific experience in literature in order to categorize participants in remembering and not remembering the pieces. These behavioral data were collected and analyzed.

Data from all indexes were converted in Z-scores employing mean and standard deviation of the baseline sentences. Statistical analysis was performed through Fisher’s exact test on behavioral data, comparing the two groups (L and NL) for each of the variable: content recall, appreciation, self-declared recognition. ANOVA test was performed for each of the neurophysiological variables (AW, CE, GSR) recorded in relation to listening to poetry pieces and considering the factors: GROUP (2 levels: L, NL), CANTICA (3 levels: Inferno, Paradiso, Purgatorio), TIME POINT (3 levels: First 10 s, Central 10 s, Final 10 s) or HALF (2 levels: First half, Second half). Correlation analyses were performed between pairs of indexes (AW, CE, GSR) for data calculating the relative activity in response to poetry, obtained through Equation (3). Logistic regression analyses were performed on data from the three considered neurophysiological indexes (AW, CE, GSR) recorded in relation to listening to poetry pieces, and employing the variable GROUP as categorical factor.

## 3. Results

Given the evidences that time is a crucial point for highlighting a differential response between experts and non-experts, a multiple approach to the collected data has been conducted. In particular, Leder and colleagues reported that, considering a 10 s exposure to visual art, experts reported higher declared understanding in comparison to non-experts [46]; therefore, such time point has been considered in Section 3.1, but extending the analysis to the first, central and last 10 s of the exposure to the poetry stimuli. In addition, since Codispoti and colleagues [61] found differences when considering the first and second half of the reaction to emotional videos, in the present study, the same approach has been applied to poetry stimuli and is reported in Section 3.2. Furthermore, we also analyzed the neurophysiological reaction to the entire stimuli, in order to test whether the length of the poetry stimuli would influence the perception by experts and non-experts; such results are reported in Section 3.3. Finally, we investigated the eventual difference between experts and non-experts concerning the relative higher activity in response to poetry or paraphrase; the results have been reported in Section 3.4.

### 3.1. Analysis of the First, Central and Final 10 s of the Stimuli

Considering the listening to the Paradiso, a significantly higher GSR level for NL in comparison to L students during the first 10 s (Chi^2^ = 4.80, *p* = 0.03) was shown. On the contrary, for the same time period, the NL group showed lower CE values than the L group (Chi^2^ = 4.24, *p* = 0.04). Any difference was found by the logistic regression analysis between the groups for the other cantiche and temporal segments and for any other index.

Concerning the GSR index (Figure 1), ANOVA analysis showed a significant interaction among the variables CANTICA×TIMEPOINT×GROUP (F(4,104) = 2.80, partial eta-squared = 0.10, *p* = 0.03). In particular, the post-hoc analysis evidenced significant differences within each group. In fact, despite Paradiso eliciting significantly higher (at least *p* < 0.05 for all the pairwise comparisons) GSR values for both groups, in comparison to the other cantiche at all the investigated time points (first, central and final 10 s), only for the NL group did the first 10 s elicit higher GSR values in comparison to the central and final 10 s temporal segments (Paradiso first 10 s vs. Paradiso central 10 s *p* < 0.01 and Paradiso first 10 s vs. Paradiso last 10 s *p* < 0.001).

### 3.2. Analysis of the First and Second Half of the Stimuli

Concerning the GSR analysis (Figure 2), the ANOVA test showed a significant interaction among the factors CANTICA×HALF×GROUP (F(2,84) = 4.80, partial eta-squared = 0.10, *p* = 0.01). The post-hoc analysis showed that, for the NL group, Paradiso (both first and second half) reported significantly higher GSR values in comparison to all the other cantica segments (*p* < 0.05 for all), while for the L group, Inferno (both first and second half) showed significantly higher GSR values in comparison to all the other cantica segments (*p* < 0.05 for all), except for the second half of the Inferno in comparison to the second half of Paradiso (*p* = 0.09).

### 3.3. Analysis of the Activity Elicited during the Listening to the Entire Cantica

Concerning the emotional reaction to the entire cantica, as suggested by the arousal level indexed by the GSR, we found a higher reaction by NL in comparison to L students for the Inferno (Chi^2^ = 5.47, *p* = 0.02) and values just missing the significance for the Purgatorio (Chi^2^ = 3.34, *p* = 0.07). On the contrary, the CE in response to the listening to the just mentioned cantica elicited lower CE levels in NL in comparison to L students: Inferno (Chi^2^ = 3.38, *p* = 0.07) and Purgatorio (Chi^2^ = 6.71, *p* = 0.01).

### 3.4. Analysis of the Relative Activity Elicited by Poetry and by the Paraphrase

Concerning the relative indexes activity in response to poetry or paraphrase versions of the cantiche Equation (3), it has been evidenced that, in comparison to the L group, the NL group showed higher relative activity in response to the poetry version of the Inferno for the GSR index (Chi^2^ = 8.92, *p* < 0.01) and of the Paradiso for the CE index (Chi^2^ = 6.33, *p* = 0.01).

The correlation analysis between the indexes for each cantica reported for the L group a significant positive correlation between AW and GSR for the Inferno (*r* = 0.48, *p* = 0.03). Furthermore, for both L and NL group, we found a significant positive correlation between AW and WL values for all the cantiche (Inferno—L: *r* = 0.67, *p* = 0.001, NL: *r* = 0.65, *p* = 0.001; Paradiso—L: *r* = 0.63, *p* < 0.01, NL: *r* = 0.83, *p* > 0.0001; Purgatorio—L: *r* = 0.73, *p* < 0.0001, NL: *r* = 0.70, *p* < 0.0001).

### 3.5. Behavioral Outcomes

Statistical analysis showed a different frequency distribution of the content recall between the two groups, higher for L group in comparison to the NL group for Inferno and Paradiso (Fisher’s exact test *p* = 0.049 and *p* < 0.001, respectively) (Figure 3 top). Moreover, concerning the appreciation of the cantica, the L group showed higher rates of participants who appreciated the cantica in comparison to NL group for Paradiso and Purgatorio (Fisher’s exact test *p* = 0.02 and *p* = 0.03, respectively) (Figure 3 center). Finally, concerning the self-declared recognition of the cantica, L students in comparison to NL students reported a larger number of participants who recognized Paradiso (Fisher’s exact test *p* = 0.02) (Figure 3 bottom).

## 4. Discussion

By looking at behavioral data, there is a very immediate suggestion that in NL students, therefore non-experts, knowledge makes them happier. In other words, in the NL but not in the L group, the appreciation/liking went hand by hand with the self-declared recognition and with the content recall, which can be considered a form of understanding and knowledge. This is in accordance with fluency theories in aesthetics, predicting higher liking linked to higher successful recognition of the stimulus and to the owning of the knowledge needed for the interpretation of the stimulus; in other words, when less effortful heuristic strategies are employed then positive affective states are elicited [82]. In fact, in the present study, the highest appreciation, self-declared recognition and content recall percentages were obtained in non-experts by Inferno stanzas, that represent the most well-known and studied verses of Divina Commedia by Italian high school students. However, for the L group, other aspects possibly explaining a less clear behavioral pattern could intervene, further investigated by the neurophysiological approach.

Main neurophysiological results of the present study were: (i) an inverse pattern for GSR and CE indexes, with an opposite trend in the two groups; (ii) a positive correlation between AW and CE. 

Concerning the first point, in comparison to the L group, the NL group showed higher emotional reaction in terms of arousal, as indexed by the GSR activity, in response to the first 10 s of Paradiso and the entire length of Inferno and Purgatorio. Such results are in accord with previous electromyography data, concerning the emotional reaction indexed by the decreased contraction of the *corrugator supercilii* in art experts in comparison to non-experts when exposed to artworks but also to the International Affective Picture System (IAPS) pictures, suggesting that art experts would in general process visual stimuli differently from non-experts [83]. The present results extend such evidence also to auditory poetry stimuli, suggesting that the emotional attenuation showed by experts could be specific to the area of expertise. This observation appears aligned to the concept of the need for the testing of effects found for the visual domain with other sensory domains, in order to proceed toward a unifying model of neuroaesthetics [64]. An alternative interpretation for higher GSR values reported in NL in comparison to the L group could be found if considering the GSR as an index of listening effort [84,85], therefore suggesting that the NL group would experience higher cognitive effort in comparison to the expert group when listening to the investigated stimuli. However, further studies are needed in order to elucidate a clear role for GSR as index of listening effort, disentangling it from its possible involvement in attentional mechanisms and as influenced by emotional aspects [84,86,87,88]. It is interesting to note that the same temporal segments reported an opposite result when analyzing the CE index, therefore, showing lower CE values in NL in comparison to L students. Vice versa, the L group reported lower GSR levels and higher CE values in comparison to the same temporal segments. Such results support the occurrence of the separation between liking/appreciation and emotional arousal, already described in the theory of emotions, describing those as composed by valence and arousal [62]. Present results are also in accord with the distinction between aesthetic emotion and aesthetic judgments, outlined as outputs of the information-processing stage model of aesthetic processing proposed by Leder and colleagues [89]. Specifically, L students appear to be less moved by poetry, but more appreciative of it, as witnessed by the higher declared appreciation of the cantica by L students in comparison to NL students, except for the Inferno, which is a very famous episode of Divina Commedia, well-known and appreciated also by non-experts, since describing the love story between two lovers (Paolo and Francesca), that as well as Lancelot and Guinevere, were experiencing a forbidden love. However, it is interesting to note that, for that episode, only in the L group, a positive correlation between GSR and AW values that considered the relative activity in response to poetry was found, suggesting an unconscious cerebral appreciation as well as arousal elicitation identifiable in experts but not in non-experts, according to the notion that appreciation of art by experts involves “cognitive mastery” [89]. Furthermore, concerning the temporal segments in which an opposite tendency between experts and non-experts was evidenced, that is the first 10 s for Paradiso and the entire length of the stimuli for Inferno and Purgatorio, it could be related to the content of the different cantica. In particular, the episode taken from Paradiso was very descriptive of the environment, while Dante was proceeding toward God’s vision, while the ones taken from Inferno and Purgatorio were focused on more narrative episodes, that were the Paolo and Francesca love story and the reprimand of Dante by Beatrice, respectively. This peculiar characteristic could account for the discrepancy in time points needed for highlighting differences between experts and non-experts, in fact according to the Vienna Integrated Model of Art Perception (VIMAP), that insists on the necessity of integrating bottom-up artwork-induced processes, with top-down mechanisms occurring within the processing experience, leading to changes in persons (e.g., to be moved or disturbed) [90]. Therefore, the interpretation given to the stimuli would take more time to be fulfilled for the narrative episodes, whilst for the Paradiso already the first 10 s was enough. It is also interesting to note that only for Paradiso was a statistically significant difference obtained between the L and NL groups in all the behavioral items investigated (recall, appreciation and self-declared recognition). This would suggest that the conscious verbal evaluation of the cantica would rely on the very first time of exposure. Such time point, 10 s, has been proved to be necessary in order to observe differences between experts and non-experts in front of abstract paintings, therefore, allowing observers to assign a meaning beyond the mere description [46]. This observation could be applied in the present study making a comparison between Paradiso and abstract artworks that share rarefied features. In addition, in the period of the first 10 s a correlation between traditional and contemporary artworks’ verbal appreciation and the neuroelectric indexes estimated during their observation was already shown [91]. Concerning the evidence that, in response to Paradiso, higher GSR responses were reported at all 10 s length time points considered in the study in both groups, it could be due to the just mentioned rarefied, sublime atmospheres of the cantica. Specifically, the fact that only in the NL group, the first 10 s showed higher GSR values in comparison to the central 10 s and the last 10 s, could be due to the fact that non-experts react more tightly basing their artworks evaluation on their gut response, consistently with the feelings-as-information theory [13]. This could also explain the fact that in the NL group, both the first and second half of the Paradiso obtained higher GSR values in comparison to all the other cantica halves. Furthermore, it could be suggested that for Paradiso stanzas, predominantly a “faster” perceptual fluency (i.e., the ease of identifying the physical features of a stimulus) would occur, while for the more narrative Purgatorio and Inferno stanzas, a “slower” conceptual fluency (i.e., the ease of mental operations concerned with stimulus meaning and its relation to semantic knowledge structures) [92] would prevail—fluency components that resemble the bottom-up and top-down processes mentioned above. However, such fluency components could be summarized with the concept of processing fluency [13], which could instead explain commonalities between L and NL groups when analyzing the entire cantiche’s excerpts, without taking account for the time points of the exposure. The correlation between AW and CE found in both groups could be discussed in line with the cognitive fluency theory [13] mentioned above. In fact, rhyme and meter, focused in the present study through the calculation of the relative neurophysiological activity in response to poetry Equation (3), can increase beauty while decreasing the ease of semantic processing (given the words choice and order constraints); however, the balance between these two components, which could be synthesized in perception versus semantics, was still positive [93]. 

The correlation found between AW and GSR only in L students in response to Inferno stanzas, when considering the relative activity in response to poetry than to paraphrase, could be explained by the peak shift effect [94]. Such effect states that when a rat is rewarded when, for instance, performing the discrimination between a rectangle (target) and a square, it will be even more responsive when facing a rectangle presenting more pronounced length differences between base and high in comparison to the prototype. Similarly, comparing the paraphrase version with the prototype and the poetry version with the more characterized geometric shape, it would be possible to explain the reason of the neurophysiological appreciation and arousal by L students but not by NL students, because of the higher capacity of attributing meaning to the poetry stimulus and semantic meaning knowledge in the formers (conceptual meaning). The reason of such reaction only to Inferno stanzas would be related to the highly emotional nature of the content, describing the passionate verses of the Paolo and Francesca love story. This could also explain, in the L group, the higher GSR (for both the first and second half of the stimuli) in comparison to almost all the other cantica halves. However, the analysis of the cantica halves did not evidence a usefulness of focusing in such a time window for the investigation of differences between groups. In fact, results did not show differences between groups due to the half portion of the cantica’s excerpts, but rather evidenced within group differences in the emotional reaction to the different Cantiche. This result appears in accord with Codispoti and colleagues [61] who did not find, between groups (men and women), differences in GSR levels in the analysis of the reaction to the first and second half of emotional movies. It is interesting to note, instead, that the same authors found an effect of the interval, not retrieved in the present study, suggesting a sensitivity of the employed time window for movie stimuli, not extendible to literary stimuli. Anyway, further research is needed in order to verify or not such suggestion. 

## 5. Conclusions

We summarize, in relation to the experimental objectives stated at the end of the introduction section:The present study evidenced that also in literary art, and in particular poetry, emotionally activating processes appeared more pronounced in non-experts (NL students in the present study) than in experts (L students). On the contrary, experts resulted to be more cognitively engaged on the same stimuli segments, further supporting the suggestion that time of exposure to an artistic stimulus constitutes a matter of importance [34,46];It was suggested that for the formation of declarative judgements (appreciation, recognition) non-experts would more strictly rely on the ease of processing, based on previous knowledge of the artistic stimuli, while experts would rely more to the cerebral processing in response to the stimuli, therefore more to top-down processes. This would be supported by the evidence that cognitive and emotional aspects are more intertwined in non-experts [83].

Future research may benefit from the combination between a systematic emotional assessment as formulated into the Aesthetic Emotions Scale (AESTHEMOS) [95] and the neurophysiological assessment, in experts and non-experts. This approach would provide further insight into the eventual relation between elements of the aesthetic experience evaluated through the AESTHEMOS scales (prototypical aesthetic emotions, epistemic emotions, and emotions indicative of amusement) and subscales.

## Figures and Tables

**Figure 1 brainsci-11-00281-f001:**
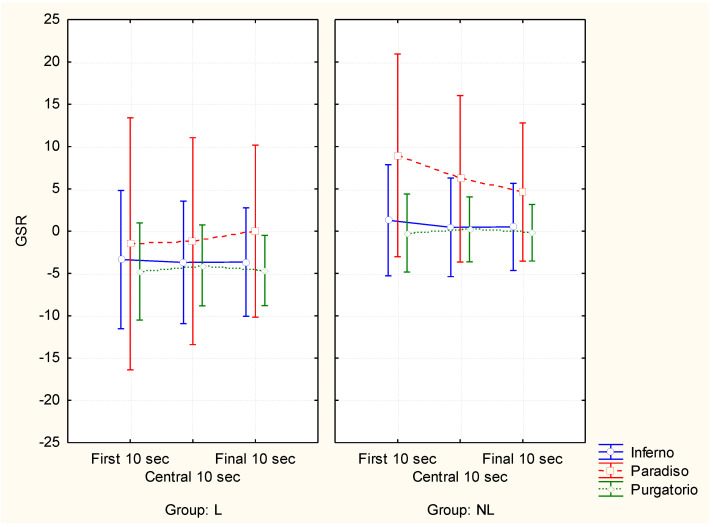
Graph representing the significant interaction between the factors: CANTICA (Inferno, Paradiso and Purgatorio), TIME POINT (first, central and final 10 s segments) and GROUP (L: Literature students; NL: Non-Literature students) resulting from the ANOVA analysis for the Galvanic Skin Response (GSR) index. Vertical bars denote 0.95 confidence interval.

**Figure 2 brainsci-11-00281-f002:**
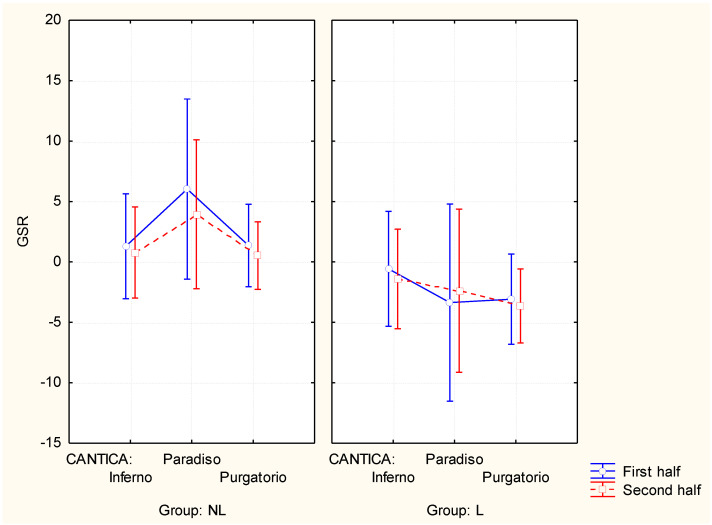
Graph representing the significant interaction between the factors: CANTICA (Inferno, Paradiso and Purgatorio), HALF (first and second half segments) and GROUP (L: Literature students; NL: Non-Literature students) resulting from the ANOVA analysis for the GSR index. Vertical bars denote 0.95 confidence interval.

**Figure 3 brainsci-11-00281-f003:**
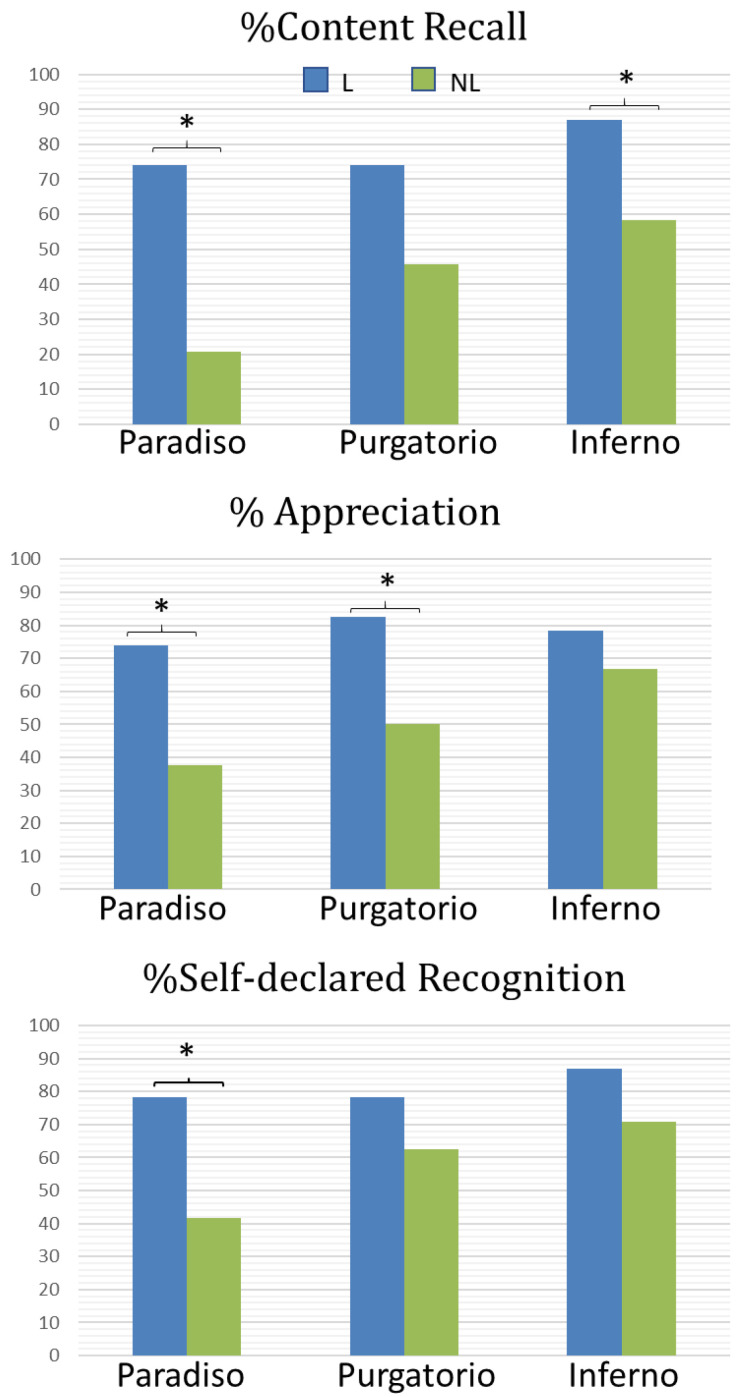
Graph representing the significant differences between the GROUPS (L: Literature students; NL: Non-Literature students) resulting from the Fisher’s exact test performed on behavioral data. Asterisks denote a statistical significance equal or lower than * *p* = 0.05.

**Table 1 brainsci-11-00281-t001:** The table summarizes mentioned cerebral processes and relative brain areas (see the main text for further details).

Cerebral Process	Brain Area	Reference
Involvement in the processing of stories with an emotional content	Medial and bilateral Orbitofrontal Cortex	[5]
Involvement in the processing of audio and visual beautiful stimuli	Medial Orbitofrontal Cortex	[6]
Higher level cognitive processing (e.g. comprehension and reasoning)	Prefrontal Cortex	[7]
Involvement in emotional words processing	Inferior Frontal Gyrus	[10]
Semantic retrieval; involvement in positive words processing	Superior frontal gyrus	[10]
Relative EEG alpha asymmetry (approach-withdrawal tendency)	Prefrontal Cortex	[14,15,16,17,18,19,20,21,22,23,24,25,26,27,28,29,30,31,32,33,34,35]
EEG correlates of expertise in response to specific stimuli and task execution (theta synchronization and or alpha desynchronization)	Frontal area	[36,37,40,41,54]
EEG correlates of expertise in response to specific stimuli and task execution (alpha synchronization)	Parietal area	[40,54]
EEG correlates of expertise in response to specific stimuli and task execution (alpha desynchronization)	Distribution over the scalp	[42]
EEG correlates of expertise in response to specific stimuli and task execution (theta and gamma synchronization) in dancers	Distribution over the scalp	[43]
EEG correlates of expertise in response to specific stimuli and task execution (alpha and beta synchronization) in musicians	Distribution over the scalp	[43]

## Data Availability

The data presented in this study are available on request from the corresponding author, without undue reservation.

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
