# Peer review of "NeuroDante: Poetry Mentally Engages More Experts but Moves More Non-Experts, and for Both the Cerebral Approach Tendency Goes Hand in Hand with the Cerebral Effort"

_brainsci, 2021, doi:10.3390/brainsci11030281_

Round 1
Reviewer 1 Report
This study investigates possible differences between expert and non-expert literature readers' auditory perception of three parts of Dante Alighieri's poem "The Divine Commedy", focusing on behavioural (recall, recognition, and appreciation) as well as physiological (electrodermal response, approach-withdrawal index, cerebral effort index) measures. This study has the potential to be of interest to readers interested in experimental approaches to poetry. However, despite my overall interest in the topic of the study and its potential, I find that the study is lacking sufficient methodological detail. I outline my main concerns below:
- The authors seem to analyse the approach/withdrawal and cerebral effort indices yet there is no mention in the article of precisely how those indices were calculated and what data were used for their calculation. Also, a shortage of prior peer-reviewed work that would implement such indices in similar experimental contexts to a significant extent questions both the validity and reliability of both metrics.
- Prior work in the field of neuropoetics (e.g. a series of studies by Hsu et al.) included a number of measures to select and control their stimuli (for a discussion of commonly used methods to investigate poetry see Jacobs, 2015, https://doi.org/10.3389/fnhum.2015.00186). In the present study, there is no information of how the three passages were controlled for syntactic and/or semantic variables. Although the researchers seem to be very interested in the perceived emotionality of the three passages by expert and nonexpert literature readers, they did not conduct a post-experimental valence and arousal ratings of those passages. Instead, the study included a yes/no question about the participants' appreciation of a given passage, which does not provide a valid and reliable index of self-reported valence and arousal.
- Because of the fact that elevated levels of GSR could index both increased cognitive effort associated with listening to a passage AND a higher emotional reaction to the passage - how do the authors disentagle those two possible interpretations? For example, how can one be sure that higher GSR values in NL participants reflect their emotional engagement rather than greater cognitive effort associated with listening to rather challenging literary text (as compared to experts)? Overall, based on the data in the study I do not find it convincing that the study provides evidence "suggesting an emotional attenuation "expertise-specific" showed by experts, but increasedd cognitive processing in response to the stimuli".
- Some of the questions asked in the study were not actually addressed or answered in the results section. For example, in section 3.2. the authors wanted to analyse the difference between the first and second half of the stimuli. However, although they report an interaction between Cantica, Half, and Group, they do not discuss how the groups and cantica differed (if at all) between the first and second half - wasn't that the purpose of this analysis? Instead the results focus solely on one side of the interaction :: the relationship between the Group (L, NL) and Cantica (Inferno, Paradiso, Purgatorio).
- My final comment relates to how the paper is written. The whole paper should be proof-read by a proficient speaker of English (or a native English speaker), because in some places it is very difficult to understand the authors' arguments, which of course impacts the reader's interpretation of those arguments.
Author Response
Reviewer 1:
1. The authors seem to analyse the approach/withdrawal and cerebral effort indices yet there is no mention in the article of precisely how those indices were calculated and what data were used for their calculation. Also, a shortage of prior peer-reviewed work that would implement such indices in similar experimental contexts to a significant extent questions both the validity and reliability of both metrics.
We thank the Reviewer for the comment, in the original version we referred to previous studies for such technical details, but we agree that for convenience it is useful to add them in the manuscript, therefore we improved the manuscript adding such information on the calculation of the indices, as follows:
“The EEG activity was recorded using a 10-electrodes-based EEG frontal band (Fpz, Fp1, Fp2, AFz, AF3, AF4, AF5, AF6, AF7, AF8) by means of a portable 24-channels system (BEmicro, EBneuro, Italy). The signals were acquired at a sampling rate of 256 Hz and the impedances were kept below 10 kΩ. In order to reject the main current interference, it was applied a notch filter (50 Hz), and then the gathered signal was digitally band-pass filtered by a 5th order Butterworth filter ([2÷30] Hz), in order to reject the continuous component as well as high-frequencies interferences, such as muscular artifacts. Successively, in order to identify and remove other artifacts-related components, such as blinks and eye movements the Independent Component Analysis (ICA) was applied to EEG data, (Di Flumeri et al., 2016a). Furthermore, in order to take into account any subjective difference in terms of brain rhythms, for each subject the Individual Alpha Frequency (IAF) was computed on the 60-s-long Closed Eyes segment [73], recorded at the beginning of the experimental task, in order to define the EEG bands of interest as: theta [IAF-6 ÷ IAF-2 Hz] and alpha [IAF-2 ÷ IAF+2 Hz] [70]. Moreover, it was calculated the Global Field Power (GFP) [74], so to summarize the activity of the cortical areas of interest in a specific frequency band. Specifically, in the present study the GFP was computed from specific set of electrodes for each index, by performing the sum of squared values of EEG potential at each electrode, averaged for the number of involved electrodes,resulting in a time-varying waveform related to the increase or decrease of the global power in the analyzed EEG. The GFP was defined according to the formula (1):
〖GFP〗_(ϑ,Frontal) (t)=1/N ∑_(i=1)^Nâ–’x_(i,ϑ) 〖(t)〗^2, (1)
where ϑ is the considered EEG band, Frontal is the considered cortical area, N is the number of electrodes included in the area of interest, and i is the electrodes’ index. Also, the GFP function was averaged on 1-s-long signal windows, so to comply with the EEG signal stationarity hypothesis [75].
In particular, for the AW index calculation was employed the following formula (2) [71]:
AW = GFPα_right − GFPα_left, (2)
where the GFPα_right and GFPalpha_left stand for the GFP calculated among right (Fp2, AF4, AF6, AF8) and left (Fp1, AF3, AF5, AF7) electrodes respectively, in the alpha (α) band. Positive AW values stand for a participant’s approach tendency, while negative AW values for a withdrawal tendency in relation to the stimulus.
Concerning the CE index, GFP in the theta band over all the frontal electrodes (Fpz, AFz, Fp2, AF4, AF6, AF8, AF7, AF3, Fp1, AF5) was considered for the index computation. Increased frontal theta (that is CE) values would imply an increase in the task difficulty [76].
Concerning the GSR, the electrodermal activity was recorded by means of a NeXus-10 (Mindmedia, The Netherlands) system with a sampling rate of 128 Hz and the skin conductance acquired by the constant voltage method (0.5 V). The electrodes were attached, on the non-dominant hand, to the palmar side of the middle phalanges of the second and third fingers of the participant, following published procedures [77]. The tonic component of the skin conductance level was obtained using LEDAlab software [78].”
Concerning previous work that employed such indices in neuropoetics, despite a wide application of them in several contexts (please see in the text e.g. G. Cartocci et al., “Frontal brain asymmetries as effective parameters to assess the quality of audiovisual stimuli perception in adult and young cochlear implant users,” Acta Otorhinolaryngologica Italica, vol. 37, pp. 1–15, 2017, doi: 10.14639/0392-100X-1407; G. Vecchiato et al., “EEG frontal asymmetry related to pleasantness of music perception in healthy children and cochlear implanted users,” in 2012 Annual International Conference of the IEEE Engineering in Medicine and Biology Society (EMBC), Aug. 2012, pp. 4740–4743, doi: 10.1109/EMBC.2012.6347026; G. Cartocci, A. G. Maglione, E. Modica, D. Rossi, P. Cherubino, and F. Babiloni, “Frontiers | Against smoking public service announcements, a neurometric evaluation of effectiveness.” http://www.frontiersin.org/10.3389/conf.fnhum.2016.220.00096/event_abstract (accessed Dec. 22, 2016); G. Cartocci et al., “Electroencephalographic, Heart Rate, and Galvanic Skin Response Assessment for an Advertising Perception Study: Application to Antismoking Public Service Announcements,” J Vis Exp, no. 126, Aug. 2017, doi: 10.3791/55872; E. Modica et al., “Neuroelectrical indices evaluation during antismoking public service announcements on a young population,” in 2017 IEEE 3rd International Forum on Research and Technologies for Society and Industry (RTSI), Sep. 2017, pp. 1–5, doi: 10.1109/RTSI.2017.8065949; E. Modica et al., “Neurophysiological profile of antismoking campaigns.”; G. Cartocci et al., “A pilot study on the neurometric evaluation of ‘effective’ and ‘ineffective’ antismoking public service announcements,” 2016, pp. 4597–4600; E. Modica et al., “Neurophysiological Responses to Different Product Experiences,” Computational Intelligence and Neuroscience, vol. 2018, 2018; G. Vecchiato et al., “Neurophysiological Tools to Investigate Consumer’s Gender Differences during the Observation of TV Commercials,” Comput Math Methods Med, vol. 2014, 2014, doi: 10.1155/2014/912981; P. Cherubino et al., “Neuroelectrical Indexes for the Study of the Efficacy of TV Advertising Stimuli,” in Selected Issues in Experimental Economics, Springer, 2016, pp. 355–371; P. Cherubino et al., “Consumer Behaviour through the Eyes of Neurophysiological Measures: State-of-the-Art and Future Trends,” Computational Intelligence and Neuroscience, vol. 2019, 2019; G. Cartocci et al., “Gender and Age Related Effects While Watching TV Advertisements: An EEG Study,” Comput Intell Neurosci, vol. 2016, 2016, doi: 10.1155/2016/3795325; G. Vecchiato et al., “Spectral EEG frontal asymmetries correlate with the experienced pleasantness of TV commercial advertisements,” Med Biol Eng Comput, vol. 49, no. 5, pp. 579–583, Feb. 2011, doi: 10.1007/s11517-011-0747-x; P. Cherubino, A. G. Maglione, I. Graziani, A. Trettel, G. Vecchiato, and F. Babiloni, “Measuring Cognitive and Emotional Processes in Retail: A Neuroscience Perspective,” Successful Technological Integration for Competitive Advantage in Retail Settings, 2015. www.igi-global.com/chapter/measuring-cognitive-and-emotional-processes-in-retail/126365 (accessed Nov. 24, 2020); F. Babiloni et al., “The great beauty: A neuroaesthetic study by neuroelectric imaging during the observation of the real Michelangelo’s Moses sculpture,” in 2014 36th Annual International Conference of the IEEE Engineering in Medicine and Biology Society, Aug. 2014, pp. 6965–6968, doi: 10.1109/EMBC.2014.6945230; F. Babiloni et al., “Neuroelectric brain imaging during a real visit of a fine arts gallery: a neuroaesthetic study of XVII century Dutch painters,” Annu Int Conf IEEE Eng Med Biol Soc, vol. 2013, pp. 6179–6182, 2013, doi: 10.1109/EMBC.2013.6610964.; F. Babiloni et al., “The first impression is what matters: a neuroaesthetic study of the cerebral perception and appreciation of paintings by Titian,” Annu Int Conf IEEE Eng Med Biol Soc, vol. 2015, pp. 7990–7993, Aug. 2015, doi: 10.1109/EMBC.2015.7320246; P. Aricò, G. Borghini, G. Di Flumeri, A. Colosimo, S. Pozzi, and F. Babiloni, “A passive brain-computer interface application for the mental workload assessment on professional air traffic controllers during realistic air traffic control tasks,” Prog. Brain Res., vol. 228, pp. 295–328, 2016, doi: 10.1016/bs.pbr.2016.04.021; G. Borghini et al., “EEG-Based Cognitive Control Behaviour Assessment: an Ecological study with Professional Air Traffic Controllers,” Sci Rep, vol. 7, Apr. 2017, doi: 10.1038/s41598-017-00633-7; Borghini, G., Isabella, R., Vecchiato, G., Toppi, J., Astolfi, L., Caltagirone, C., & Babiloni, F. (2011). Brainshield: HREEG study of perceived pilot mental workload. Italian journal of aerospace medicine, 5(1), 34-47; M. Doppelmayr, T. Finkenzeller, and P. Sauseng, “Frontal midline theta in the pre-shot phase of rifle shooting: differences between experts and novices,” Neuropsychologia, vol. 46, no. 5, pp. 1463–1467, Apr. 2008, doi: 10.1016/j.neuropsychologia.2007.12.026) supporting their validity and reliability, to our best knowledge their application to literary and in particular poetry stimuli is quite novel, being already employed only in the following articles respectively:
Approach/Withdrawal (Frontal Alpha Asymmetry):
- -M. Brouwer, M. Hogervorst, B. Reuderink, Y. van der Werf, and J. van Erp, “Physiological signals distinguish between reading emotional and non-emotional sections in a novel,” Brain-Computer Interfaces, vol. 2, no. 2–3, pp. 76–89, Apr. 2015, doi: 10.1080/2326263X.2015.1100037.
- Cartocci et al., “The ‘NeuroDante Project’: Neurometric Measurements of Participant’s Reaction to Literary Auditory Stimuli from Dante’s ‘Divina Commedia,’” in Symbiotic Interaction, Sep. 2016, pp. 52–64, doi: 10.1007/978-3-319-57753-1_5.
Cerebral Effort (Frontal Theta):
- Cartocci et al., “The ‘NeuroDante Project’: Neurometric Measurements of Participant’s Reaction to Literary Auditory Stimuli from Dante’s ‘Divina Commedia,’” in Symbiotic Interaction, Sep. 2016, pp. 52–64, doi: 10.1007/978-3-319-57753-1_5.
2. Prior work in the field of neuropoetics (e.g. a series of studies by Hsu et al.) included a number of measures to select and control their stimuli (for a discussion of commonly used methods to investigate poetry see Jacobs, 2015, https://doi.org/10.3389/fnhum.2015.00186). In the present study, there is no information of how the three passages were controlled for syntactic and/or semantic variables. Although the researchers seem to be very interested in the perceived emotionality of the three passages by expert and nonexpert literature readers, they did not conduct a post-experimental valence and arousal ratings of those passages. Instead, the study included a yes/no question about the participants' appreciation of a given passage, which does not provide a valid and reliable index of self-reported valence and arousal.
We thank the reviewer for such worthy indications. Concerning Hsu’s studies, in the introduction, we added to the text the following references: Hsu CT, Jacobs AM, Altmann U, Conrad M. The magical activation of left amygdala when reading Harry Potter: an fMRI study on how descriptions of supra-natural events entertain and enchant. PLoS One. 2015 Feb 11;10(2):e0118179. doi: 10.1371/journal.pone.0118179. PMID: 25671315; PMCID: PMC4324997; Hsu CT, Jacobs AM, Citron FM, Conrad M. The emotion potential of words and passages in reading Harry Potter--an fMRI study. Brain Lang. 2015 Mar;142:96-114. doi: 10.1016/j.bandl.2015.01.011. Epub 2015 Feb 11. PMID: 25681681.
We agree about the importance of methods in investigating poetry, in fact already in the original manuscript version we cited the suggested article by Jabos (2015), according to which “Following Jakobson, literary texts can methodically be described by their (1) metric; (2) phonological; (3) syntactic; or (4) semantic properties (and others, of course)”. Accordingly, we tested the passages accounting for the first of the mentioned items, being the metric and the structure of Divina Commedia extremely constant. In addition, being the passages belonging to the same narrative framework, that is the Divina Commedia, a certain grade of semantic and syntactic consistency is preserved (Canettieri P. Unified Theory of the Text (UTT) and the Question of Authorship Attribution, in «Memoria di Shakespeare», n.s. 8, 2012, pp. 65-77). However, we can see that a focus on additional criteria would be of interest, despite beyond the purposes of the present study, but for sure object of further investigations.
Concerning the second point of the comment, we totally agree that a traditional practice for checking for the perceived valence and arousal of stimuli is through rating of such dimensions by participants, for instance on a Likert scale. However, since the widely famous nature of Divina Commedia among Italians, we were worried about a possible bias in detailed valence and arousal ratings, due to social desiderability (e.g. Britz, S., Gauggel, S. & Mainz, V. The Aachen List of Trait Words. J Psycholinguist Res 48, 1111–1132 (2019). https://doi.org/10.1007/s10936-019-09649-8). Therefore we preferred to investigate through indices already correlated in scientific literaure to valence (i.e. approach/withdrawal) (Harmon-Jones E, Gable PA. On the role of asymmetric frontal cortical activity in approach and withdrawal motivation: An updated review of the evidence. Psychophysiology. 2018 Jan;55(1). doi: 10.1111/psyp.12879. Epub 2017 Apr 29. PMID: 28459501) and arousal (GSR) (M. M. Bradley, M. Codispoti, B. N. Cuthbert, and P. J. Lang, “Emotion and motivation I: defensive and appetitive reactions in picture processing,” Emotion, vol. 1, no. 3, pp. 276–298, Sep. 2001; P. J. Lang and M. M. Bradley, “Emotion and the motivational brain,” Biol Psychol, vol. 84, no. 3, pp. 437–450, Jul. 2010, doi: 10.1016/j.biopsycho.2009.10.007; M. Codispoti, P. Surcinelli, and B. Baldaro, “Watching emotional movies: Affective reactions and gender differences,” International Journal of Psychophysiology, vol. 69, no. 2, pp. 90–95, Aug. 2008, doi: 10.1016/j.ijpsycho.2008.03.004).
3. Because of the fact that elevated levels of GSR could index both increased cognitive effort associated with listening to a passage AND a higher emotional reaction to the passage - how do the authors disentagle those two possible interpretations? For example, how can one be sure that higher GSR values in NL participants reflect their emotional engagement rather than greater cognitive effort associated with listening to rather challenging literary text (as compared to experts)? Overall, based on the data in the study I do not find it convincing that the study provides evidence "suggesting an emotional attenuation "expertise-specific" showed by experts, but increasedd cognitive processing in response to the stimuli".
We thank the Reviewer for such important question and interesting alternative interpretation of results. According to Pichora-Fuller and colleagues (Pichora-Fuller, M. K., Kramer, S. E., Eckert, M. A., Edwards, B., Hornsby, B. W., Humes, L. E., ... & Wingfield, A. (2016). Hearing impairment and cognitive energy: The framework for understanding effortful listening (FUEL). Ear and hearing, 37, 5S-27S), GSR activity is mediated by the sympathetic nervous system and is considered a measure of cognitive (and in particular of listening) effort, despite a not very wide application of it for such kind of estimation (e.g. Mackersie, C. L., & Calderon-Moultrie, N. (2016). Autonomic nervous system reactivity during speech recognition tasks: Heart-rate variability and skin conductance. Ear Hear, 37, 118S–125S). Furthermore, according to Francis and colleagues (Francis, A. L., MacPherson, M. K., Chandrasekaran, B., & Alvar, A. M. (2016). Autonomic nervous system responses during perception of masked speech may reflect constructs other than subjective listening effort. Frontiers in psychology, 7, 263) electrodermal response does not perfectly match with performances and perceived difficulty, in fact authors found differences in electrodermal responses between two challenging listening conditions, despite performance and listeners’ subjective perception of task demand were comparable across the conditions. Moreover, authors hypothesize a role for skin conductance response in attentional mechanisms but expressing the need of further studies to prove or not such hypothesis, moreover overly complicating the role of GSR during effortful listening scenarios. Finally, Alhanbali and colleagues showed a lack in test-retest reliability for GSR as index of listening effort, whilst authors found that skin conductance was more sensitive to emotional factors than the other measures of listening effort (i.e. EEG and pupillometry) used in the study (Alhanbali S, Dawes P, Millman RE, Munro KJ. Measures of Listening Effort Are Multidimensional. Ear Hear. 2019 Sep/Oct;40(5):1084-1097. doi: 10.1097/AUD.0000000000000697. PMID: 30747742; PMCID: PMC7664710.).
The above considerations would support the higher sensitivity of GSR as an emotional dimension index.
However, accordingly to Reviewer’s comment we decided to include in the manuscript the possible interpretation suggested by the Reviewer, since the presence in scientific literature of suggestions in this sense, as follows:
“An alternative interpretation for higher GSR values reported in NL in comparison to L group, could be find if considering the GSR as an index of listening effort [83], [84], therefore suggesting that NL group would experience higher cognitive effort in comparison to the expert group when listening to the investigated stimuli. However further studies are needed in order to elucidate a clear role for GSR as index of listening effort, disentangling it from its possible involvement in attentional mechanisms and influenced by emotional aspects [83], [85]–[87].”
4. Some of the questions asked in the study were not actually addressed or answered in the results section. For example, in section 3.2. the authors wanted to analyse the difference between the first and second half of the stimuli. However, although they report an interaction between Cantica, Half, and Group, they do not discuss how the groups and cantica differed (if at all) between the first and second half - wasn't that the purpose of this analysis? Instead the results focus solely on one side of the interaction: the relationship between the Group (L, NL) and Cantica (Inferno, Paradiso, Purgatorio).
We thank the Reviewer for the comment and apologise for the missing information in the original version of the manuscript. We added Accordingly, since no differences between groups were found in the analysis of the Cantica halves, we specified in the Discussion section the following sentences:
“This could also explain, in the L group, the higher GSR (for both the first and second half of the stimuli) in comparison to almost all the other Cantica halves. However, the analysis of the Cantica halves did not evidence a usefulness of focusing on such time-window for the investigation of differences between groups. In fact, results did not show differences between groups due to the half portion of the Cantica’s excerpts, but rather evidenced within group differences in the emotional reaction to the different Cantica. This result appears in accord to Codispoti and colleagues [59] who did not find between group (men and women) differences in GSR levels in the analysis of the reaction to the first and second half of emotional movies. It is interesting to note instead that the same authors found an effect of the interval, not retrieved in the present study, suggesting a sensitivity of the employed time-window for movie stimuli, not extendible to literary stimuli. Anyway, further research is needed in order to verify or not such suggestion.”
The lack of statistically significant effect of the factors GROUP and CANTICA when considering the first and second half of the passages, could be
5. My final comment relates to how the paper is written. The whole paper should be proof-read by a proficient speaker of English (or a native English speaker), because in some places it is very difficult to understand the authors' arguments, which of course impacts the reader's interpretation of those arguments.
We thank the Reviewer and apologise for the lack of clarity. Accordingly, we tried to improve the English level of the manuscript.

Reviewer 2 Report
The manuscript by Cartocci et al., is a relatively comprehensive research article on the potential effects of literature on the behavior outcome. The authors focus on the potential behavior differences in groups regarding aesthetics. Investigations are well performed. There are some mild-to-moderate concerns:
- In Section Introduction, Lines 43-44, “orbitofrontal cortex” and “social cognition” both mentioned, but not tested in the authors’ research.
- Lines 48-90 and related summary, please briefly select the neural mechanisms (including different brain regions) for an additional table.
- Experimental design for Section Materials and Methods, Lines 142-155, gender/sex differences not addressed. To improve the quality of research, the data is needed.
- Section 5 Behavior outcomes, please provide all F-factor. In addition, because of limited n number, power analysis can be performed. The current statistics is not acceptable.
- Line 300 and related experimental design, missing important descriptions on “neurophysiological” tests.
Author Response
Reviewer 2:
- In Section Introduction, Lines 43-44, “orbitofrontal cortex” and “social cognition” both mentioned, but not tested in the authors’ research.
We thank the Reviewer for the comment and apologise for the lack of clarity in the manuscript. The orbitofrontal cortex (OFC), according to the International 10-20 System, correspond to the EEG electrode positions Fpz, Fp1 and Fp2 (Bognár, A., Csete, G., Németh, M., Csibri, P., Kincses, T. Z., & Sáry, G. (2017). Transcranial Stimulation of the Orbitofrontal Cortex Affects Decisions about Magnocellular Optimized Stimuli. Frontiers in neuroscience, 11, 234. https://doi.org/10.3389/fnins.2017.00234; Megan L. Willis, Jillian M. Murphy, Nicole J. Ridley, Ans Vercammen, Anodal tDCS targeting the right orbitofrontal cortex enhances facial expression recognition, Social Cognitive and Affective Neuroscience, Volume 10, Issue 12, December 2015, Pages 1677–1683, https://doi.org/10.1093/scan/nsv057). We therefore included those electrodes into our investigation, but extending the area of analysis to a wider area of the prefrontal cortex, since involved in higher-level cognitive processes, like comprehension and reasoning, known to be crucial for texts processing (I. Goethals, K. Audenaert, C. Van de Wiele, and R. Dierckx, “The prefrontal cortex: insights from functional neuroimaging using cognitive activation tasks,” Eur J Nucl Med Mol Imaging, vol. 31, no. 3, pp. 408–416, Mar. 2004, doi: 10.1007/s00259-003-1382-z).
Our mentioning of the OFC was functional to the explanation of the interest and reliability of researches focused on neurocognitive poetics, but our aim was not to focus on such specific brain area, nor to investigate particular social cognition aspects in the present research. Our aim in the lines highlighted by the Reviewer was just to provide to the reader background concerning brain areas already investigated in scientific literature and proved to be involved during exposure to literary material characterized by emotional content, in order to explicate the contextualization of the research in relation to the special issue matter.
- Lines 48-90 and related summary, please briefly select the neural mechanisms (including different brain regions) for an additional table.
We thank the Reviewer for the valuable suggestion, accordingly we added the following table:
- Experimental design for Section Materials and Methods, Lines 142-155, gender/sex differences not addressed. To improve the quality of research, the data is needed.
We thank the Reviewer for highlighting such missing information and apologise for it. We added it into the Materials and Methods section. In particular, the L group was composed by 12 F and 11 M, while the NL group by 12 F and 12 M. Therefore it is possible to see that a strong gender balance was respected between the groups.
- Section 5 Behavior outcomes, please provide all F-factor. In addition, because of limited n number, power analysis can be performed. The current statistics is not acceptable.
We thank the Reviewer for the comment and apologise for the lack of clarity. For the analysis of the behavioural outcomes, as the considered variables were categorical (e.g. recall: yes/no), in order to investigate differences between the groups, we employed the non-parametric Fisher’s exact test calculated on Statistica software, and such test does not provide F values, but only the two-tailed P-value, that we reported in the manuscript. In addition, concerning the number of included participants, the Fisher’s exact test is suitable for small sample size (Camilli, G., & Hopkins, K. D. (1979). Testing for association in 2 × 2 contingency tables with very small sample sizes. Psychological Bulletin, 86(5), 1011–1014. https://doi.org/10.1037/0033-2909.86.5.1011).
- Line 300 and related experimental design, missing important descriptions on “neurophysiological” tests.
We thank the Reviewer for the comment and apologise for the lack of clarity. We added to the manuscript detailed information about the neurophysiological tests and indices, as follows:
“The EEG activity was recorded using a 10-electrodes-based EEG frontal band (Fpz, Fp1, Fp2, AFz, AF3, AF4, AF5, AF6, AF7, AF8) by means of a portable 24-channels system (BEmicro, EBneuro, Italy). The signals were acquired at a sampling rate of 256 Hz and the impedances were kept below 10 kΩ. In order to reject the main current interference, it was applied a notch filter (50 Hz), and then the gathered signal was digitally band-pass filtered by a 5th order Butterworth filter ([2÷30] Hz), in order to reject the continuous component as well as high-frequencies interferences, such as muscular artifacts. Successively, in order to identify and remove other artifacts-related components, such as blinks and eye movements the Independent Component Analysis (ICA) was applied to EEG data, (Di Flumeri et al., 2016a). Furthermore, in order to take into account any subjective difference in terms of brain rhythms, for each subject the Individual Alpha Frequency (IAF) was computed on the 60-s-long Closed Eyes segment [73], recorded at the beginning of the experimental task, in order to define the EEG bands of interest as: theta [IAF-6 ÷ IAF-2 Hz] and alpha [IAF-2 ÷ IAF+2 Hz] [70]. Moreover, it was calculated the Global Field Power (GFP) [74], so to summarize the activity of the cortical areas of interest in a specific frequency band. Specifically, in the present study the GFP was computed from specific set of electrodes for each index, by performing the sum of squared values of EEG potential at each electrode, averaged for the number of involved electrodes,resulting in a time-varying waveform related to the increase or decrease of the global power in the analyzed EEG. The GFP was defined according to the formula (1):
〖GFP〗_(ϑ,Frontal) (t)=1/N ∑_(i=1)^Nâ–’x_(i,ϑ) 〖(t)〗^2, (1)
where ϑ is the considered EEG band, Frontal is the considered cortical area, N is the number of electrodes included in the area of interest, and i is the electrodes’ index. Also, the GFP function was averaged on 1-s-long signal windows, so to comply with the EEG signal stationarity hypothesis [75].
In particular, for the AW index calculation was employed the following formula (2) [71]:
AW = GFPα_right − GFPα_left (2)
where the GFPα_right and GFPalpha_left stand for the GFP calculated among right (Fp2, AF4, AF6, AF8) and left (Fp1, AF3, AF5, AF7) electrodes respectively, in the alpha (α) band. Positive AW values stand for a participant’s approach tendency, while negative AW values for a withdrawal tendency in relation to the stimulus.
Concerning the CE index, GFP in the theta band over all the frontal electrodes (Fpz, AFz, Fp2, AF4, AF6, AF8, AF7, AF3, Fp1, AF5) was considered for the index computation. Increased frontal theta (that is CE) values would imply an increase in the task difficulty [76].
Concerning the GSR, the electrodermal activity was recorded by means of a NeXus-10 (Mindmedia, The Netherlands) system with a sampling rate of 128 Hz and the skin conductance acquired by the constant voltage method (0.5 V). The electrodes were attached, on the non-dominant hand, to the palmar side of the middle phalanges of the second and third fingers of the participant, following published procedures [77]. The tonic component of the skin conductance level was obtained using LEDAlab software [78].”

Round 2
Reviewer 1 Report
I would like to thank the authors for addressing my concerns. I think that the manuscript has improved significantly relative to the original submission. Specifically, adding the section on EEG data acquisition and preprocessing improved the clarity of the methods used in the study. I would only suggest to include this description under a different heading (e.g. for EEG : "Electrophysiological data acquisition and preprocessing"; for SCR" "Electrodermal data acquisition and preprocessing"). My final comment concerns the language of the article that - to my mind - should still be improved. I think it is in the authors' best interest to make their paper communicate clearly what they want to say. Hence, I would recommend that prior to publication the paper be proofread by a native English speaker.
Author Response
We thank the Reviewer for the suggestion, we feel to have improved the English quality of the article.
Reviewer 2 Report
Revision has been well performed and previous concerns have been well addressed. The manuscript has been significantly improved. Minor points: Table 1 needs rearrangement for the first and third columns. Line 440 “thanks to the calculation” was unclear. Future research also needs to address gender differences.
Author Response
We thank the Reviewer for the comment. We changed the expression at Line 440 to "through the calculation ". Concerning the rearrangement of Table 1, for convenience we maintained the arrangement according to the order of appearance in the manuscript of the relative references; in addition we corrected a typo. Finally, we totally agree with the Reviewer, in future studies we will investigate also the effect of the gender on such topic.